

# Global evaluation of Doppler velocity errors of EarthCARE Cloud Profiling Radar using global storm-resolving simulation

Yuichiro Hagihara[1], Yuichi Ohno[1], Hiroaki Horie[1], Woosub Roh[2], Masaki Satoh[2], and Takuji Kubota[3]

[1]Radio Research Institute, National Institute of Information and Communications Technology, Koganei, Tokyo 184-8795, Japan
[2]Atmosphere and Ocean Research Institute, The University of Tokyo, Kashiwa, Chiba 277-8564, Japan
[3]Earth Observation Research Center, Japan Aerospace Exploration Agency, Tsukuba, Ibaraki 305-8505, Japan

**Correspondence**: Yuichiro Hagihara (hagihara@nict.go.jp)

**Abstract.** The Cloud Profiling Radar (CPR) on the Earth Clouds, Aerosol, and Radiation Explorer (EarthCARE) satellite is the first satellite-borne Doppler radar (EC-CPR). In our previous study, we examined the effects of horizontal (along-track) integration and simple unfolding methods on the reduction of Doppler errors in the EC-CPR observations, and those effects were evaluated using two limited scenes in limited latitude and low pulse repetition frequency (PRF) settings. In this study, the amount of data used was significantly increased, and the area of the data used was extended globally. Not only low PRF but also high PRF settings were examined. We calculated the EC-CPR-observed Doppler velocity from pulse-pair covariances using the radar reflectivity factor and Doppler velocity obtained from a satellite data simulator and a global storm-resolving simulation. The global data were divided into five latitudinal zones, and mean Doppler errors for 5 dBZ$_e$ after 10 km integration were calculated. In the case of low PRF setting, the error without unfolding correction for the tropics reached a maximum of 2.2 m s$^{-1}$ and then decreased toward the poles (0.43 m s$^{-1}$). The error with unfolding correction for the tropics became much smaller at 0.63 m s$^{-1}$. In the case of high PRF setting, the error without unfolding correction for the tropics reached a maximum of 0.78 m s$^{-1}$ and then decreased toward the poles (0.19 m s$^{-1}$). The error with unfolding correction for the tropics was 0.29 m s$^{-1}$, less than half the value without the correction. The results of the analyses of the simulated data indicated that the zonal mean frequency of precipitation echoes was highest in the tropics and decreased toward the poles. Considering a limitation of the unfolding correction for discrimination between large upward velocity and large precipitation falling velocity, the latitudinal variation of the Doppler error can be explained by the precipitation echo distribution.

## 1 Introduction

The Earth Clouds, Aerosol and Radiation Explorer (EarthCARE; hereafter EC) is a joint satellite mission by the Japan Aerospace Exploration Agency (JAXA) and European Space Agency (ESA) that will carry a Cloud Profiling Radar (CPR), an ATmospheric LIDar (ATLID), a Multi Spectral Imager (MSI), and a Broad Band Radiometer (BBR). From the derived 3D cloud and aerosol scene profiles, heating rates and radiation flux profiles are systematically determined with a resolution of



100 km$^2$ (Illingworth et al., 2015). Active sensors of EC will be regarded as an evolutional successor of the 94-GHz CloudSat CPR (Stephens et al., 2008) and the Cloud-Aerosol Lidar and Infrared Pathfinder Satellite Observations (CALIPSO; Winker et al., 2009) lidar (Stephens et al., 2018).

Because of EC's low orbit (~400 km) and the EC-CPR's large antenna (2.5 m), it has a better sensitivity (–36 dBZ$_e$ at the top-of-atmosphere (TOA)) than the CloudSat CPR (–30 dBZ$_e$) and can observe 98 % of radiatively significant ice clouds and 40 % of all stratocumulus clouds (Stephens et al., 2002; Hagihara et al., 2010). Moreover, the EC-CPR has the vertical Doppler measurement capability that the CloudSat CPR does not have. It will reveal, for the first time, the vertical motion of cloud particles globally. Such an entirely new dataset would improve the discrimination between clouds and precipitation (Ceccaldi et al., 2013; Kikuchi et al., 2017), as well as the retrieval of cloud microphysical parameters (Heymsfield et al., 2008).

Consequently, it should improve various parameterization schemes used in atmospheric models and the understanding of the processes related to cloud and precipitation (Roh and Satoh, 2014; Roh et al., 2017; Roh and Satoh, 2018; Hagihara et al., 2014; Mülmenstädt et al., 2020; Takahashi et al., 2021).

    Vertical Doppler velocity estimation from space suffers from Doppler broadening and velocity folding or aliasing (e.g., Kobayashi et al., 2002; Sy et al., 2014). Hagihara et al. (2022; hereafter, H22) examined the effect of horizontal (along-track)

integration and unfolding methods on the reduction of Doppler velocity measurement errors, in order to improve Doppler data processing in the JAXA standard algorithm. They obtained EC-CPR data simulated by a satellite data simulator, the Joint-Simulator (Hashino et al., 2016; Satoh et al., 2016; Roh et al., 2020) using a global storm-resolving simulation data with the Nonhydrostatic ICosahedral Atmospheric Model (NICAM; Tomita and Satoh, 2004; Satoh et al., 2008; Satoh et al., 2014). They evaluated the Doppler errors for each $Z_e$ for two cases (cirrus clouds and precipitation). They found that the error was

reduced by horizontal integration alone in the case of cirrus clouds, whereas the error became large without unfolding correction in addition to the horizontal integration in the case of precipitation.

    In H22, the evaluation was limited to two scenes in the mid-latitudes of the Northern Hemisphere and a low pulse repetition frequency (PRF) setting. In this study, we used more data than in H22 and performed the evaluation on a global scale. We also adopted different PRF settings. In Sect. 2, the simulation methods for EC-CPR data, the horizontal integration and unfolding

correction of Doppler velocity, and the CloudSat-observed data are described. In Sect. 3, we investigate the Doppler errors on a global scale. To examine the characteristics of each latitude, we separated the data into five latitudinal zones. Two PRF modes were also included. The summary and conclusions are given in Sect. 4.

## 2 Data and Method

We utilized the global storm-resolving simulation data simulated by the NICAM with a 3.5 km horizontal resolution. Moreover,

we obtained the simulated EC-CPR data using the data and the Joint-Simulator following H22. Note that attenuations of the gas and particle are considered in the calculation of the radar reflectivity factor, whereas Doppler velocity is the total velocity of the hydrometer echo, including reflectivity-weighted particle fall speed and vertical air motion. Those





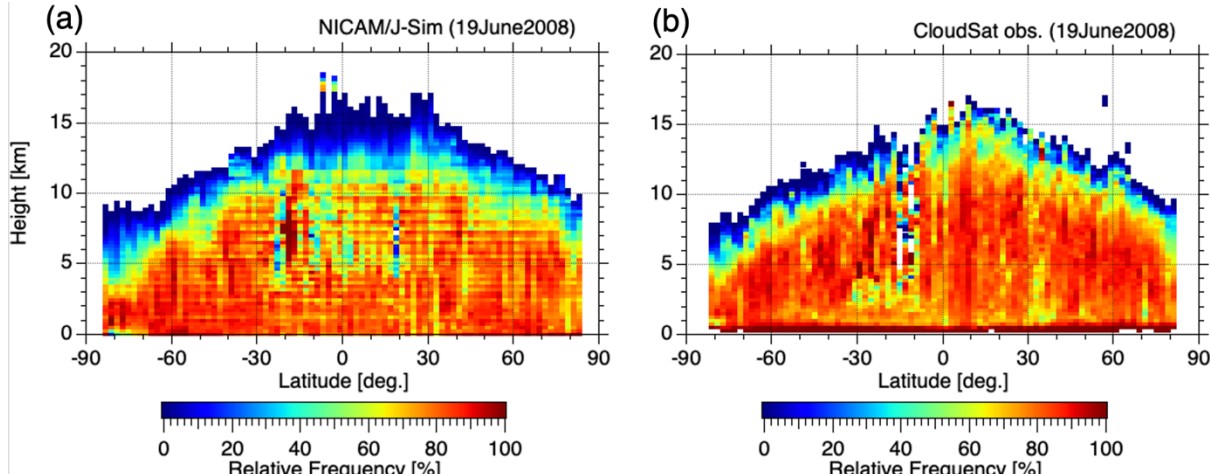

**Figure 1.** Zonal mean frequency of hydrometeors obtained by (a) NICAM/J-Sim and (b) CloudSat observations for 19 June 2008.


data were then calculated along an EC orbit and interpolated into the EC-CPR sampling interval (100 m in vertical and 500 m in horizontal). The radar reflectivity factor ($Z_{e, jsim}$) and Doppler velocity ($V_{jsim}$) curtain data were obtained (hereinafter referred to as "NICAM/J-Sim data"). In H22, only two scenes extracted from two orbits of data were used, but in this study, the amount of data used was significantly increased to 16 orbits of data, which is equivalent to one day of satellite tracks.

In using the NICAM/J-Sim data, we first performed the following statistical analyses. We examined the zonal mean frequencies of hydrometeors obtained from the NICAM/J-Sim data and the CloudSat observations for 19 June 2008 (Fig. 1). We used the CloudSat $Z_e$ (the standard geometrical profile of cloud product, 2B-GEOPROF) (Stephens et al., 2008) for comparison with $Z_{e, jsim}$. For the observed data, we used the CPR Level 2B-GEOPROF cloud mask product to extract bins with threshold ≥20 that are less affected by surface clutter and other factors. These are estimated to have a false-positive probability

of 5 % (Marchand et al., 2008). The frequency at a given altitude was defined as the number of cloud echo bins ($Z_e > -24$ dB$Z_e$) divided by the total number of observations at that level. The bin size was 240 m in vertical and 2.0° latitude in horizontal. The overall frequencies of the NICAM/J-Sim simulated cloud field are comparable to the results of the CloudSat observations.

We simulated the measured vertical Doppler velocity ($V_m$) as

$$V_m = V_{jsim} + V_{random},\qquad(1)$$

where $V_{random}$ is the random error caused by the spread of Doppler velocities within the beam width. This is a Gaussian error distribution, and its SD of random error ($SD_{random}$) is determined by perturbation approximation (Doviak and Zrnic, 1993) as





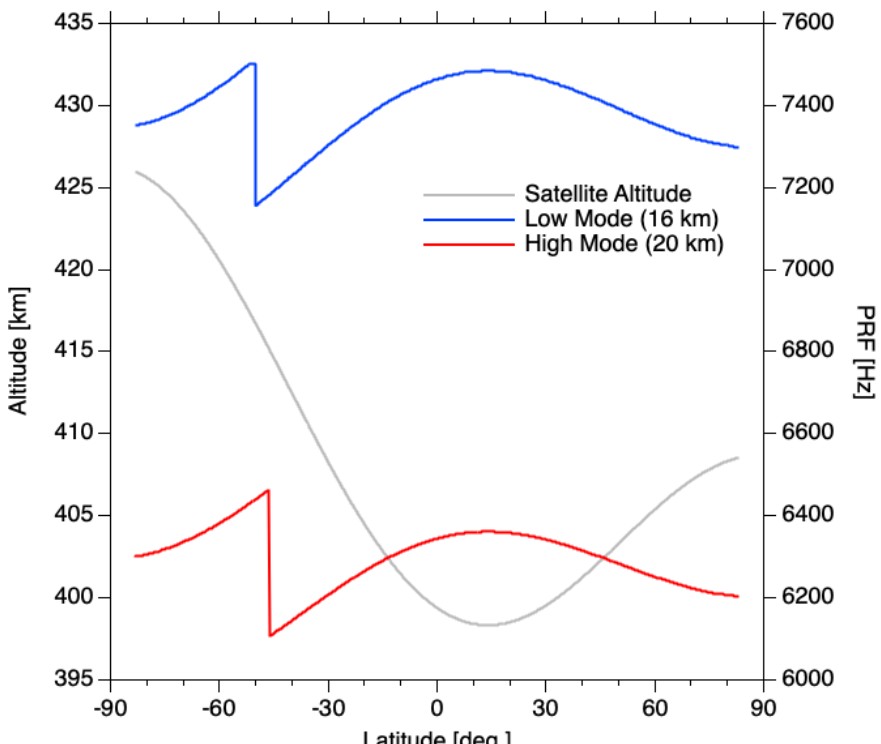

**Figure 2.** Satellite altitude and PRF as functions of latitude and observation mode.


$$SD_{random} = C \sqrt{\frac{\lambda^2}{32\pi^2 M \cdot \rho^2 \cdot \left(\frac{1}{PRF}\right)^2}\left[\left(1+\frac{N}{S}\right)^2 - \rho^2\right]}, \qquad (2)$$

and $C$ is a correction factor. We set $C = 1.3$ following H22. The wavelength is $\lambda$ ($\lambda = 3.2$ mm for EC-CPR), $M$ is the number of pulse pairs within an integration length, $\rho$ is the correlation function, and $S/N$ is the SNR. In nominal operation, the EC-CPR will change the observation window, that is, low mode (–1 to 16 km altitude) at latitudes of 60 to 90° and high mode (–1 to 20 km) at latitudes of 0 to 60°. The PRF is determined on the basis of the satellite altitude and changes in the range of 6100 to 7500 Hz with the latitude and observation window, as illustrated in Fig. 2. The high mode has a lower PRF and worse Doppler accuracy, as discussed in H22, although cloud echoes up to an altitude of 20 km can be observed. On the other hand, the low mode has a higher PRF and better Doppler accuracy, but cloud echoes higher than 16 km cannot be observed. $M$ is 357 to 420 for 500 m integration depending on the PRF. The SNR is determined by the received echo power calculated from the radar equation and estimated EC-CPR noise level. In the case of EC-CPR, the SNR is 0 dB, which is a signal equivalent to –21.2 dB$Z_e$ echo intensity. If $Z_{e, jsim}$ is less than –24 dB$Z_e$, we assume the Doppler velocity of its echo to be random noise in this study. The correlation function $\rho$ is defined as





$$\rho = exp\left\{-8\left(\frac{\pi \cdot \sigma_v}{\lambda \cdot PRF}\right)^2\right\}, \tag{3}$$

where $\sigma_v$ is the total Doppler velocity spectrum width. In this study, we assumed $\sigma_v = 4.01$ m s$^{-1}$.

The EC-CPR measures the phase change of the echo between two successive pulses by pulse-pair processing to estimate the Doppler velocities. The real and imaginary parts of pulse-pair covariances $R_\tau$ integrated onboard corresponding to a 500 m along-track are simulated in this study as

$$Re\left(R_\tau\right) = Z_{e,jsim} \cdot cos\left(\frac{4\pi \cdot V_m}{\lambda \cdot PRF}\right), \tag{4}$$

$$Im\left(R_\tau\right) = Z_{e,jsim} \cdot sin\left(\frac{4\pi \cdot V_m}{\lambda \cdot PRF}\right). \tag{5}$$

$V_{500m}$ is calculated using the arctangent of the real and imaginary parts of the 500-m-integrated $R_\tau$ simulated by Eqs. (4) and (5). The sign of Doppler velocity is defined as being those of radial Doppler velocity (i.e., downward motion is positive) following the EC-CPR data processing. To reduce random error, $V_{1km}$ and $V_{10km}$ are also calculated using 1 and 10 km horizontally integrated $R_\tau$ respectively, that are calculated from the 500 m-integrated $R_\tau$.

Velocity folding or aliasing is inherent to Doppler radar. $V_{max}$ can be measured by the pulse-pair method and is defined by

PRF ($V_{max} = \lambda \cdot PRF/4$). In the high-mode PRF, $V_{max}$ ranges from 4.9 to 5.2 m s$^{-1}$, whereas in the low-mode PRF, it ranges from 5.7 to 6.0 m s$^{-1}$.

The simulated EC-CPR Doppler velocities are required for unfolding correction. To correct the velocity folding in space-borne radar, it is difficult to use the conventional unfolding method generally used by ground-based Doppler weather radar (e.g., Bargen and Brown, 1980). From the ground-based vertically pointing cloud radar observations (Horie et al., 2000),

upward motion above 3 m s$^{-1}$ was rarely observed. On the basis of this, we thus assumed that the echoes with velocities higher than 3 m s$^{-1}$ are upward folded precipitation echoes. We used the simple unfolding method as follows:

$$V_{unfolded} = \begin{cases} V_{folded} + 2 \cdot V_{max} & \text{for } V_{1km,10km} < -3m/s \\ V_{folded} & otherwise. \end{cases} \tag{6}$$

## 3 Results

We first evaluated the global mean Doppler errors in the high-mode PRF as well as low-mode PRF. Then, we separated the

NICAM/J-Sim data into five latitudinal zones (Arctic, Northern midlatitude, tropics, Southern midlatitude, and Antarctic). The Doppler errors for each latitudinal zone are investigated in both PRF modes.

Figure 3 shows the global mean Doppler errors in the high-mode PRF. The vertical axis indicates the SD of Doppler error that is calculated from the difference between the simulated velocity (i.e., $V_{1km}$, $V_{10km}$) and $V_{jsim}$ (hereafter, $SD_{diff}$). The horizontal axis indicates $Z_e$ of the NICAM/J-Sim data. The red dashed lines show $SD_{diff}$ and the solid lines indicate $SD_{diff}$ with

unfolding correction using Eq. (6). Figure 3a shows $SD_{diff}$ of $V_{1km}$ and $SD_{diff}$ of $V_{1km}$ with unfolding correction. $SD_{diff}$ of $V_{1km}$ decreases for $Z_e$ below $-10$ dBZ$_e$. This is attributed to the reduction of random error owing to the increase in $S/N$ and



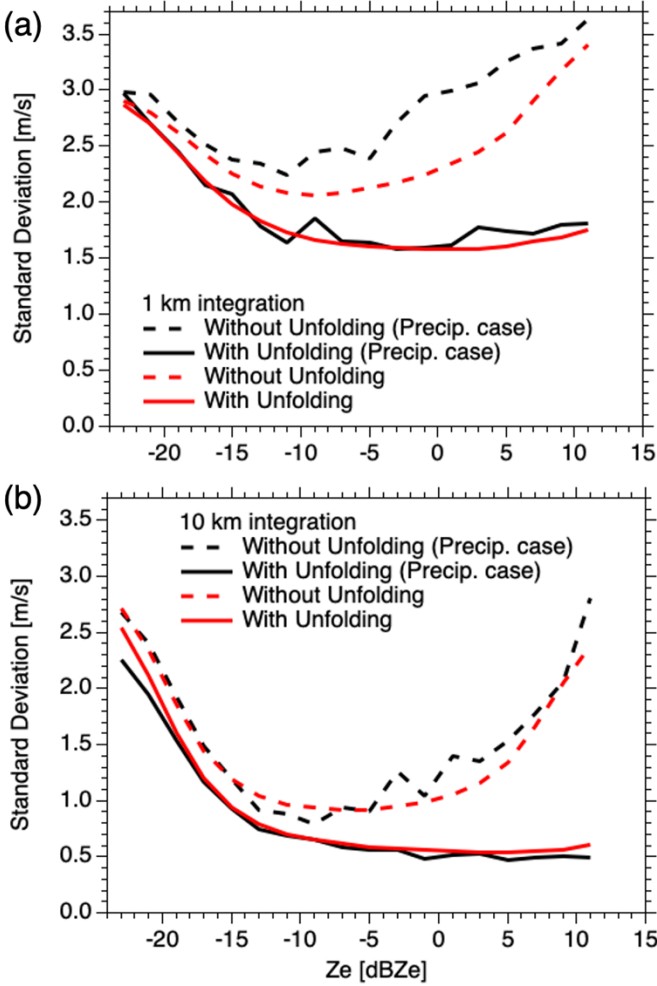

**Figure 3.** Standard deviation of random error of simulated Doppler velocities for high-mode PRF as a function of $Z_e$ for (a) 1 km integration and (b) 10 km integration. The solid lines denote the results with unfolding correction. The black lines indicate the precipitation case in Hagihara et al. (2022).


decrease in $SD_{random}$ in Eq. (2) as $Z_e$ increases. $SD_{diff}$ of $V_{1km}$ increases for $Z_e$ above –10 dB$Z_e$. This is due to the increase in the occurrence of velocity folding. That is, an increase in $Z_e$ results in an increase in the intensity of precipitation echoes and an increase in mean fall velocity. When the unfolding method is applied, $SD_{diff}$ of $V_{1km}$ is noticeably reduced because the folded negative velocities are corrected and the occurrence of the velocity folding is reduced. In Fig. 3b, $SD_{diff}$ of $V_{10km}$ decreases for

$Z_e$ below –7 dB$Z_e$ and increases for $Z_e$ above –7 dB$Z_e$. $SD_{diff}$ of $V_{10km}$ is much smaller than that of $V_{1km}$, reaching 0.8 m s$^{-1}$ for –9 dB$Z_e$. This is because of the increase in $M$ and the decrease in $SD_{random}$ in Eq. (2). If the unfolding method is applied, $SD_{diff}$ of $V_{10km}$ becomes smaller since the effect of folding Doppler errors of precipitation echoes is reduced, as shown in Fig. 3a. For instance, $SD_{diff}$ of $V_{10km}$ is less than 0.5 m s$^{-1}$ above –5 dB$Z_e$.



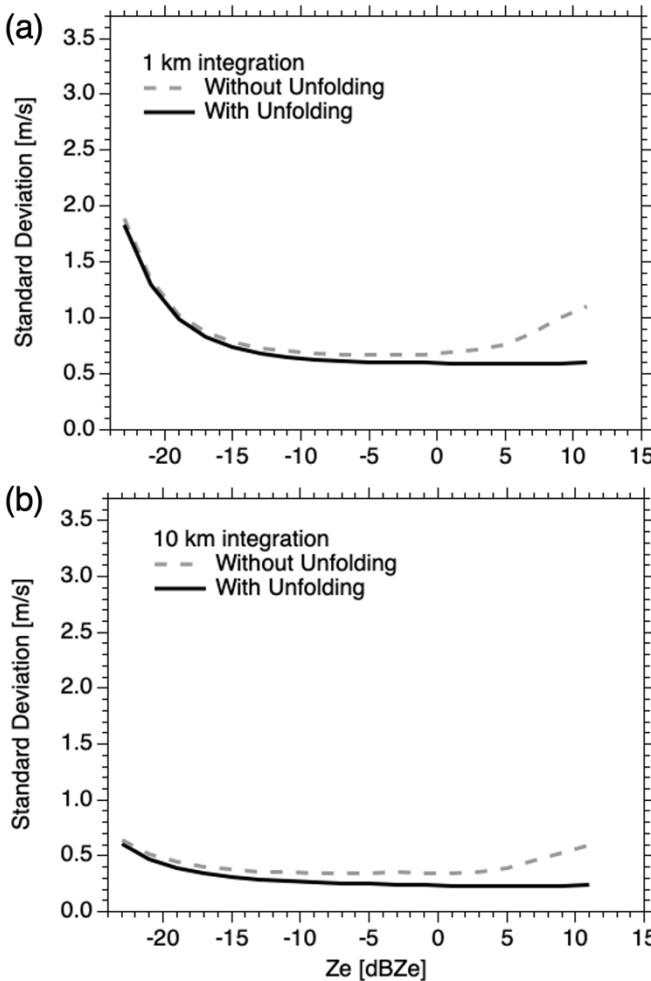

**Figure 4.** Standard deviation of random error of simulated Doppler velocities for low-mode PRF as a function of $Z_e$ for (a) 1 km integration and (b) 10 km integration. The solid lines denote the results with unfolding correction.

What has been described so far is consistent with what was shown in the analysis of the precipitation case in H22. Note that PRF varied from 6106 to 6464 Hz in the high mode illustrated in Fig. 2 but was a single value of 6279 Hz in the precipitation case in H22. Incidentally, we added black dashed and solid lines to Fig. 3 to show the results. In both Figs. 3a and 3b, the results are in good agreement with those of this study.

Figure 4 illustrates the global mean Doppler errors in the low-mode PRF. The dashed lines show $SD_{diff}$ without unfolding correction and the solid lines indicate $SD_{diff}$ with unfolding correction using Eq. (6). The PRF varies from 7156 to 7500 Hz, with a corresponding $SD_{random}$ of 1.5 to 3.4 for 0 to –19 dBZ$_e$ (see Fig. 2 in H22). On the other hand, in the high mode, the PRF varies from 6106 to 6464 Hz, with a corresponding $SD_{random}$ of 0.8 to 1.5 for 0 to –19 dBZ$_e$. Similarly, $V_{max}$ takes values between 5.7 and 6.0 m s$^{-1}$, whereas in the high mode, it is between 4.9 and 5.2 m s$^{-1}$. Comparison of Figs. 3 and 4 clearly shows that



the Doppler error is much smaller in the latter because of $SD_{random}$ described above. Furthermore, $SD_{diff}$ without unfolding correction is smaller than that in the high-mode PRF because $V_{max}$ is larger in addition to the effect of $SD_{random}$.

Since the frequencies of cloud and precipitation echoes differ in latitude and the PRF varies with latitude, as shown in Fig.
2, we investigated the change in $SD_{diff}$ with latitude. We defined five latitudinal zones, namely, Arctic (>60°), Northern midlatitude (60° to 30°), tropics (30° to –30°), Southern midlatitude (–30° to –60°), and Antarctic (<–60°). In the following analysis, we focused on $SD_{diff}$ of $V_{10km}$. Figs. 5a–5e show the Doppler error for the five latitudinal zones in the high-mode PRF. The dashed lines show $SD_{diff}$ without unfolding correction and the solid lines indicate $SD_{diff}$ with unfolding correction using Eq. (6). $SD_{diff}$ of $V_{10km}$ without unfolding correction decreases up to a certain value of $Z_e$ and increases after that value. $SD_{diff}$
with unfolding correction decreases as $Z_e$ increases. These tendencies observed in the five latitudinal zones are similar to those of the global mean $SD_{diff}$ of $V_{10km}$ shown in Fig. 3b, although their magnitudes are not the same. We compared $SD_{diff}$ without unfolding correction. $SD_{diff}$ for the tropics, shown in Fig. 5c, has the largest value and is larger than the global mean result. The $SD_{diff}$ values for both midlatitudes (Figs. 5b and 5d) are smaller than that for the tropics but slightly larger than or comparable to the global mean result. The $SD_{diff}$ values for both polar regions (Figs. 5a and 5e) are even smaller than those for
both midlatitudes and smaller than the global mean result. $SD_{diff}$ for the Antarctic in Fig. 5e shows the smallest value. The tendency of the magnitude relation of $SD_{diff}$ for each latitudinal zone was the similar between without and with unfolding correction. From the PRF variation shown in Fig. 2, the Doppler accuracy should be higher in the tropics and lower toward the poles. However, the results we have seen so far are opposite. On the other hand, the frequency of precipitation echoes is considered to be the highest in the tropics, and the resulting folding Doppler error may have resulted in the largest $SD_{diff}$ being
in the tropics.

Figures. 5f–5j demonstrate the Doppler error for the five latitudinal zones in the low-mode PRF. The dashed lines show $SD_{diff}$ without unfolding correction and the solid lines indicate $SD_{diff}$ with unfolding correction using Eq. (6). Similarly to Figs. 3 and 4, comparison of Figs. 5a–5e and 5f–-j shows that $SD_{diff}$ is much smaller in the latter. There is a difference between with and without unfolding correction only for $SD_{diff}$ for the tropics shown in Fig. 5h, but not for the others. This may be related to
the frequency of precipitation echoes, as also explained in Figs. 5a–5e. In the low-mode PRF, $V_{max}$ is larger and $SD_{random}$ is smaller owing to the higher PRF.

To summarize what has been discussed so far, the $SD_{diff}$ values for the five latitudinal zones for 5 $dBZ_e$ were extracted and shown in Fig. 6. The red crosses indicate $SD_{diff}$ without unfolding correction of the high-mode PRF, and the red circles denote $SD_{diff}$ with unfolding correction using Eq. (6). The red dashed line is $SD_{diff}$ for 5 $dBZ_e$ without unfolding correction, and the
red solid line is that with unfolding correction shown in Fig. 3b. $SD_{diff}$ without unfolding correction (red crosses) for the tropics is the largest at 2.2 m s$^{-1}$ and decreases in both polar directions, with the smallest value at 0.43 m s$^{-1}$ in the Antarctic. The $SD_{diff}$ values for the Northern midlatitude and Arctic are slightly larger than those for the Southern midlatitude and Antarctic. In comparison with the global mean $SD_{diff}$ without unfolding correction, the values for the tropics and

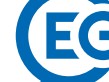



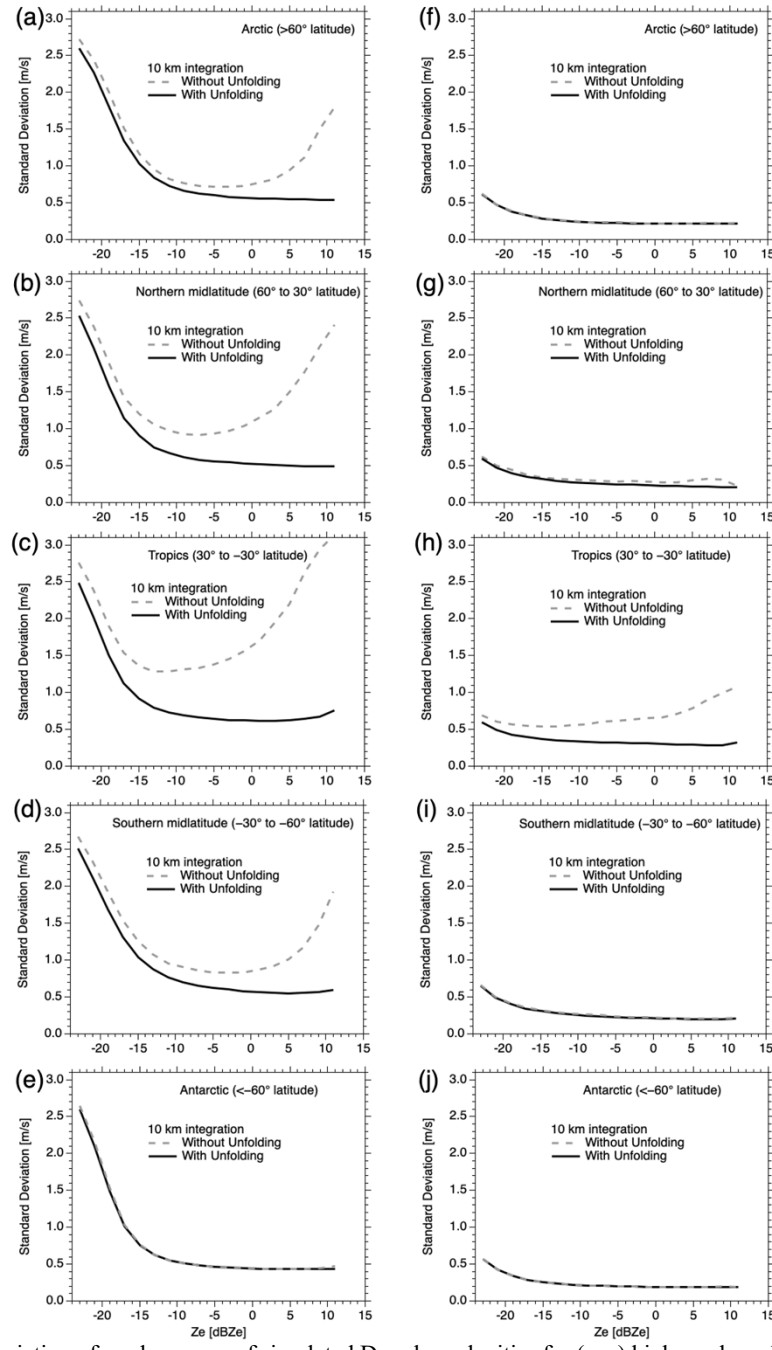

**Figure 5.** Standard deviation of random error of simulated Doppler velocities for (a–e) high-mode and (f–i) low-mode PRF as a function of $Z_e$ after 10 km integration for (a, f) Arctic, (b, g) Northern midlatitude, (c, h) tropics, (d, i) Southern midlatitude, and (e, j) Antarctic zones. The solid lines denote the results with unfolding correction.





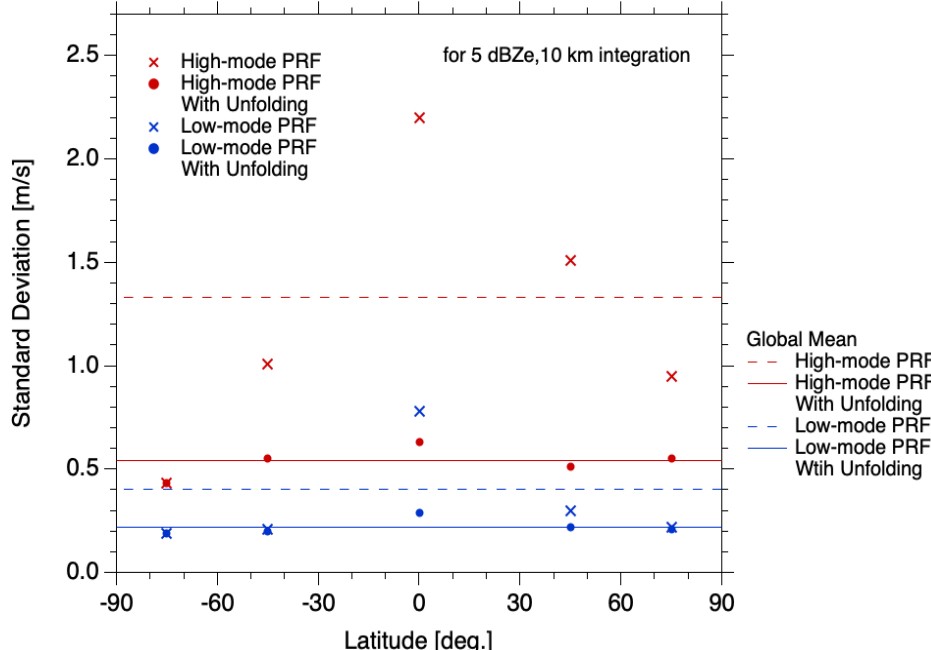

**Figure 6.** Standard deviation of random error of Doppler velocities with and without unfolding correction for 5 dB$Z_e$ after 10 km integration as a function of latitude.


Northern midlatitude are larger, but the other values are smaller. $SD_{diff}$ with unfolding correction (red circles) for the tropics is 0.63 m s$^{-1}$, which is above the global mean result of 0.54 m s$^{-1}$ in Fig. 3b. The $SD_{diff}$ values with unfolding correction for the Southern midlatitude, Northern midlatitude, and Arctic are comparable to the global mean result, but the value for the Antarctic is smaller than the global mean result. Next, we examine the low-mode PRF results. The blue crosses indicate $SD_{diff}$ without

unfolding correction of the low-mode PRF, and the blue circles denote $SD_{diff}$ with unfolding correction using Eq. (6). The blue dashed line is $SD_{diff}$ for 5 dB$Z_e$ without unfolding correction, and blue solid line is the value with unfolding correction illustrated in Fig. 4b. $SD_{diff}$ without unfolding correction (blue crosses) for the tropics is the largest at 0.78 m s$^{-1}$ and decreases toward the poles, with the smallest value being 0.19 m s$^{-1}$ at the Antarctic. $SD_{diff}$ with unfolding correction (blue circles) for the tropics is 0.29 m s$^{-1}$, which is above the global mean of 0.22 m s$^{-1}$ in Fig. 4b. The $SD_{diff}$ values with unfolding correction for the other

zones are comparable to the global mean result. As already explained in Figs. 5, the latitudinal variation of $SD_{diff}$ without unfolding correction may be due to the frequency of precipitation echoes. If the unfolding correction were perfect, there would be no relationship between the latitudinal variation of $SD_{diff}$ with unfolding correction and the frequency of precipitation echoes. However, there is actually a relationship between the two, which indicates a limitation of the unfolding correction.




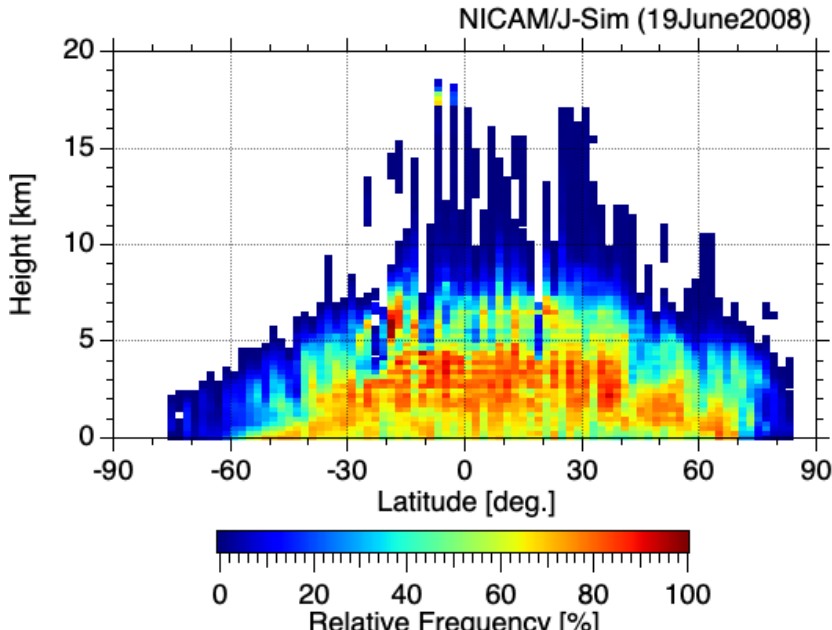

**Figure 7.** Zonal mean frequency of precipitation echoes obtained by NICAM/J-Sim for 19 June 2008.

We examined the zonal mean frequencies of precipitation echoes obtained from the NICAM/J-Sim data for 19 June 2008.

First, to obtain precipitation echoes, we used the same method as in Fig. 1a but added a Doppler velocity condition ($V_{\text{jsim}} > 3$ m s$^{-1}$, downward motion). Then, using the same bin size as in Fig. 1a, we obtained Fig. 7. The extracted precipitation echoes show that the frequency decreases at higher altitudes compared with that shown in Fig. 1a. The frequency is high in the tropics and decreases toward the poles. The frequencies at altitudes of less than 5 km were averaged by latitudinal zone and found to be as follows: 27.8 % in the Arctic, 60.3 % in the Northern midlatitude, 68.5 % in the tropics, 36.7 % in the Southern

midlatitude, and 2.6 % in the Antarctic. This is because it was summer in the Northern Hemisphere in the simulation. The latitudinal variation of $SD_{\text{diff}}$ described so far can be explained on the basis of the precipitation echo distribution.

**4 Conclusions**

We examined the vertical Doppler velocity error due to Doppler broadening and velocity folding in the EarthCARE CPR (EC-CPR) observations throughout the globe. We used simulated observation data (NICAM/J-Sim $Z_{e,\text{ jsim}}$ and $V_{\text{jsim}}$) for 16 satellite

orbits with the same sampling interval as the EC-CPR, obtained using the NICAM and a satellite data simulator, the Joint-Simulator. The EC-CPR observed 500 m horizontally integrated pulse-pair covariances and Doppler velocity. The 1 and 10 km horizontally integrated Doppler velocities were calculated from them. We evaluated the Doppler error, i.e., the standard deviation of random error ($SD_{\text{diff}}$), and investigated the effectiveness of error reduction by horizontal integration. We also evaluated the Doppler folding error by comparing the corrected Doppler velocities using our simple unfolding method.



We first evaluated the global mean Doppler error in high-mode PRF as well as low-mode PRF and compared the results with those of our previous study. In the high-mode PRF, $SD_{\text{diff}}$ without unfolding correction for 1 km integration decreases up to a certain value of $Z_e$ and increases after that value. This decreasing feature is due to the decrease in the SD of random error as the SNR increases, and the increasing feature is the result of an increase in the frequency of the folded Doppler error of precipitation echoes. $SD_{\text{diff}}$ without unfolding correction is much smaller for 10 km integration than for 1 km integration,

because of the increased number of pulse pairs. When the unfolding correction is applied, $SD_{\text{diff}}$ becomes considerably smaller regardless of the integration length and the PRF mode. The results of low-mode PRF (higher PRF) show very small Doppler error both without and with unfolding correction.

To investigate the latitudinal variation of Doppler error, we separated the data into five latitudinal zones, namely, Arctic (>60°), Northern midlatitude (60° – 30°), tropics (30° to –30°), Southern midlatitude (–30° to –60°), and Antarctic (<–60°). In

the present work, we focused on $SD_{\text{diff}}$ for 10 km integration. In the high-mode PRF, $SD_{\text{diff}}$ for the tropics without unfolding correction is the largest and is larger than the global mean result. $SD_{\text{diff}}$ without unfolding correction decreases toward the poles with the smallest value for the Antarctic, which is smaller than the global mean. The tendency of the magnitude relation of $SD_{\text{diff}}$ for each latitudinal zone was similar between without and with unfolding correction. The frequency of precipitation echoes is expected to be highest in the tropics, and the folding Doppler error is also likely to be the largest. Therefore, $SD_{\text{diff}}$

for the tropics without unfolding correction is considered to be the largest. $SD_{\text{diff}}$ is much smaller in the low-mode PRF than in the high-mode PRF, as shown by the global mean results described earlier.

In summary, $SD_{\text{diff}}$ for the five latitudinal zones for 5 dB$Z_e$ is described as follows. In the high-mode PRF, $SD_{\text{diff}}$ without unfolding correction for the tropics reached a maximum of 2.2 m s$^{-1}$ and then decreased toward the poles. $SD_{\text{diff}}$ with unfolding correction for the tropics was much smaller at 0.63 m s$^{-1}$. In the low-mode PRF, $SD_{\text{diff}}$ without unfolding correction for the

tropics reached a maximum of 0.78 m s$^{-1}$ and then decreased toward the poles. $SD_{\text{diff}}$ with unfolding correction for the tropics was 0.29 m s$^{-1}$, which is less than half the value without correction. As explained previously, the latitudinal variation of $SD_{\text{diff}}$ can be attributed to the frequency of precipitation echoes. The zonal mean frequency of precipitation echoes obtained from the NICAM/J-Sim data was higher in the tropics and decreased toward the poles. Therefore, the latitudinal variation of $SD_{\text{diff}}$ can be explained on the basis of the precipitation echo distribution.

We found that the Doppler error was higher in the tropics than in the other latitudes. In the tropics, the unfolding correction reduced the large Doppler errors more efficiently. However, there is also a limitation of the unfolding correction for discrimination between large upward velocity and large precipitation falling velocity. Comparison of the results of the low-mode and high-mode PRF settings showed that the Doppler error for the low-mode PRF setting was significantly reduced, although cloud echoes for altitudes higher than 16 km cannot be observed.



*Data availability*

CloudSat CPR data are available from the CloudSat data processing center (http://www.cloudsat. cira.colostate.edu/, last access: 12 November 2022). We can share the NICAM/J-Sim data. Please send the email (ws-roh@aori.u-tokyo.ac.jp) if interested.

*Author contributions*

Y.H. performed the data analysis and produced the final manuscript draft. Y.O. provided feedback on the analysis methods as well as on the manuscript draft. H.H. developed and maintained the algorithm code and provided feedback on the manuscript draft. W.R. preformed the Joint-Simulator simulations and provided feedback on the manuscript draft. M.S. led the NICAM development. T.K. led the Joint-Simulator development and provided feedback on the manuscript draft.

*Competing interests*

The authors declare that they have no conflict of interest.

*Acknowledgements*

The authors would like to thank the members of the JAXA EarthCARE Science Team and CPR project. The computational resources were partly provided by the National Institute for Environmental Studies.

*Financial support*

This work has been supported by the National Institute of Information and Communications Technology. This research is part of the EarthCARE satellite study commissioned by the Japan Aerospace Exploration Agency, "Enhancement of the Joint Simulator for Satellite Sensors". MS and WR were supported by the Grant-in-Aid for Scientific Research B (20H01967) and Program for Promoting Technological Development of Transportation of the Ministry of Land, Infrastructure, Transport and

Tourism (MLIT).

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
