# Peer review of "Global evaluation of Doppler velocity errors of EarthCARE Cloud Profiling Radar using global storm-resolving simulation"

_EGUsphere, 2022_

## Referee Comment (RC2)

[referee-annotated manuscript omitted]

---

## Author Comment (AC2)

Dear referee,

We really appreciate the reviewer's efforts. Your comments definitely help us to improve our paper.

The following is a point-by-point response to the specific comments:

Major comments:
1. The authors assume a constant Doppler spectrum width of 4 m/s. It would be beneficial for readers to explain why this value is a reasonable estimate given the spacecraft speed and the beam-width of the radar. Moreover, it should be also mentioned that the Doppler spectrum width depends on the observed hydrometeors distribution, i.e., for heavy rain the width will be additionally increased.

RESPONSE:
We have omitted this explanation in this paper because we have mentioned it in detail in a previous paper. However, we admit that it would certainly be more helpful to reiterate it as follows:
The width $\sigma_v$ can be considered as a sum of contributions by each. That is,

$$\sigma_v^2 = \sigma_{sm}^2 + \sigma_t^2 + \sigma_{psd}^2,$$

where $\sigma_{sm}$ is the spread due to satellite motion, given by $\sigma_{sm} \sim 0.3 V_{sat} \theta_{3dB}$, $V_{sat}$ is the satellite velocity, and $\theta_{3dB}$ is the beam width (Sloss and Atlas 1968). When $V_{sat}$ is 7738 m/s and $\theta_{3dB}$ is 0.00166 rad (0.095°), $\sigma_{sm}$ becomes 3.85 m/s. The spread $\sigma_t$ is due to turbulence and $\sigma_{psd}$ to the distributions of hydrometeor falling velocities, respectively, which are assumed to be $\sigma_t = 1.0$ m/s (Amayenc et al., 1993), and $\sigma_{psd} = 0.5$ m/s (Gossard et al., 1997). As for the latter term, it is reported to spread to 1.0 m/s for rain (Lhermitte 1963). In this study, we assumed the $\sigma_{psd,} = 0.5$ m/s so that $\sigma_v$ becomes 4.01 m/s.
We will add our explanation in the revised manuscript.

2. Although, a long integration path (10 km) seems to be a tempting approach to reduce uncertainty in the Doppler measurements, the authors do not assess the effect of a long scale signal decorrelation and non-uniform beam filling effects in such a large sampling volume. Moreover, the horizontal resolution of 10 km prevents studies on the small scale features like localized convection, the characterization of which is one of the objectives of space-borne Doppler radar missions.

RESPONSE:
We have already evaluated the change in Doppler error for 500 m, 1 km, and 10 km horizontal integrations in H22. It has been demonstrated that the Doppler error was significantly reduced by the 10-km integration. It indicated that the reduction of random errors by the integration had a larger contribution compared to the error by small-scale Doppler changes. Therefore, this paper mainly investigates the results of the 10-km integration.

3. Does the radar simulator accounts for multiple scattering? This issue has been demonstrated to have a destructive effect on the quality of the Doppler measurements.

RESPONSE:
We thank for your important remarks. However, the Doppler effect of multiple scattering was not considered in this study because of its complication, and the issue will be the subject of future research.

Minor comments:
1. The terminology "high-mode PRF" can be misleading. It would be better to use "low PRF mode" or "high tropopause mode".

RESPONSE:
We agree with your suggestion. It is indeed confusing, so we will change them in the revised manuscript as follows:
"high mode PRF" to "PRF of the high mode (lower PRF)" and
"low mode PRF" to "PRF of the low mode (higher PRF)".

2. It feels like some of the discussions can be reduced in length, e.g., when the high PRF mode is compared with the low PRF mode.

RESPONSE:
We thank you for your remarks. The main progress of this paper is that we extended the analysis to 16 orbits and also performed low mode PRF analysis and compared the results with those of high mode PRF. For the latter, we first discuss the high mode PRF results, and in the sections where we discuss the comparison with low mode PRF, we have tried to avoid redundancy and only mention what is necessary. If you feel that the discussion is redundant, it may be due to the difficulty of reading owing to the terminology problem mentioned above. We will change them in the revised manuscript in accordance with the points mentioned above, and we believe that the difficulty of reading has been improved.

Some minor comments are also included in the attached file.

Page 1,
Line 16, "mean Doppler errors for 5 dB$Z_e$":
Not clear what does it mean

RESPONSE:
Here, we mean the average Doppler errors with an echo intensity of 5dB$Z_e$.

Page 3,
Line 74 "threshold ≥20":
Is this a threshold on the reflectivity or on the cloud mask product? in the latter case, what does this number correspond to?

RESPONSE:
This is the threshold for the cloud mask product and corresponds to the "weak echo" in this product (Marchand et al., 2008). This threshold is used in many other CloudSat-based hydrometeor studies (e.g., Sassen & Wang 2008, Mace et al. 2009).

Line 75: insert "of cloud occurrence"

RESPONSE:
We will change our statement in the revised manuscript accordingly.

Page 7,
Line 145: "Incidentally, we added black dashed and solid lines to Fig. 3 to show the results":
Not clear. These figures have 4 lines that were described before. Please, rephrase this sentence.

RESPONSE:
We will correct in the revised manuscript as follows:
(before)
Incidentally, we added black dashed and solid lines to Fig. 3 to show the results.

(after)
Note the black dashed and solid lines in Fig. 3.

Line 146: insert "in H22"?

RESPONSE:
We will also change our statement in the revised manuscript as follows:
(before)
In both Figs. 3a and 3b, the results are in good agreement with those of this study.
(after)
In both Figs. 3a and 3b, the results in H22 are in good agreement with those of this study.

Line 148: "The PRF varies from 7156 to 7500 Hz, with a corresponding $SD_{random}$ of 1.5 to 3.4 for 0 to $-19$ $dBZ_e$ (see Fig. 2 in H22). On the other hand, in the high mode, the PRF varies from

6106 to 6464 Hz, with a corresponding $SD_{random}$ of 0.8 to 1.5 for 0 to –19 dB$Z_e$":
SD_random is lower for higher PRF.

RESPONSE:
We will correct in the revised manuscript as follows:
(before)
The PRF varies from 7156 to 7500 Hz, with a corresponding $SD_{random}$ of 1.5 to 3.4 for 0 to –19 dB$Z_e$ (see Fig. 2 in H22). On the other hand, in the high mode, the PRF varies from 6106 to 6464 Hz, with a corresponding SDrandom of 0.8 to 1.5 for 0 to –19 dB$Z_e$.

(after)
The PRF varies from 7156 to 7500 Hz, with a corresponding $SD_{random}$ of 0.8 to 1.5 for 0 to –19 dB$Z_e$ (see Fig. 2 in H22). On the other hand, in the high mode, the PRF varies from 6106 to 6464 Hz, with a corresponding SDrandom of 1.5 to 3.4 for 0 to –19 dB$Z_e$.

Page 8,
Line 167: "the Doppler accuracy should be higher in the tropics and lower toward the poles.":
In the tropics, high mode will be used (according to your description in line 89) that is characterized by lower PRF thus higher SD_random which contradicts what is written here.

RESONSE:
The statement in line 89 merely states what mode is used in nominal operation. However, for clarity, we will change our statement in the revised manuscript as follows:
(before)
From the PRF variation shown in Fig. 2, the Doppler accuracy should be higher in the tropics and lower toward the poles.

(after)
From the PRF variation shown in Fig. 2, in the PRF of the high mode (lower PRF), the Doppler accuracy should be higher in the tropics and lower toward the poles.

Line 169: "the frequency of precipitation echoes is considered to be the highest in the tropics, and the resulting folding Doppler error may have resulted in the largest $SD_{diff}$ being in the tropics.":
please rephrase.

RESPONSE:
We will correct our statement in the revised manuscript as follows:
(before)
the frequency of precipitation echoes is considered to be the highest in the tropics, and the resulting folding Doppler error may have resulted in the largest $SD_{diff}$ being in the tropics.

(after)
the frequency of precipitation echoes is considered to be the highest in the tropics, and the folding Doppler error may have resulted in the largest $SD_{diff}$ in the tropics.

Line 174: "This may be related to the frequency of precipitation echoes":
Rather than the frequency, it is the intensity of the precipitation that matters. The strongest radar signal is expected to be observed for the most intense rain events that are characterized by the largest raindrops thus the largest terminal and Doppler velocities. Of course, the attenuation due to rain will result in a steep reflectivity decrease toward the ground but this does not change the fact that velocity folding occurs for the most intense rain.

RESPONSE:
In H22, Fig. 9(d) shows a 2D-histogram of the Doppler velocities after 10 km integration as a function of the $Z_e$. The velocity folding occurs even for 0 dB$Z_e$, indicating that the heavy rain with large $Z_e$ does not make a significant contribution. Therefore, we consider the frequency of precipitation to be more important statistically.

Line 177 & 179: "for 5 dB$Z_e$":
Not clear what do you mean by 5 dBZe. Does it correspond to the reflectivity that exceeds 5dBZ?

RESPONSE:
Here, we mean the $SD_{diff}$ with an echo intensity of 5 dB$Z_e$.

Page 9,
According to your description, high-PRF mode will be used for latitudes beyond 60 degrees and low-PRF mode for altitudes below 60 deg. Therefore, half of the presented panels are not necessary. Please consider removing them or make a note in the figure caption which mode is expected to be used in which zone.

RESPONSE:
As already mentioned, the description in line 89 only states what mode is used in nominal operation.

Page 10, "for 5 dB$Z_e$":
Figure 6, caption & Line 196: Not clear what do you mean by 5 dBZe. Does it correspond to the reflectivity that exceeds 5dBZ?

RESPONSE:
Here, we mean the values with an echo intensity of 5 dB$Z_e$.

Line 201: "due to the frequency of precipitation echoes":
As previously, I don't think it is the frequency of occurrence that matters but the rainfall intensity.

RESPONSE:
As stated in the reply to Line 174, we consider the frequency of precipitation to be more important statistically.

Line 201: "there would be no relationship between the latitudinal variation of $SD_{diff}$ with unfolding correction and the frequency of precipitation echoes":
That is true only if only high SNR data are considered. The SD_diff depends on the SNR, thus the regions where weak radar echoes are observed often are characterized by higher SD_diff.

RESPONSE:
The latitudinal variation of $SD_{diff}$ with unfolding correction for -19dB$Z_e$ with relatively low SNR is shown below. As in fig. 6, there is no significant variation in latitude.

[Figure]

Line 242: "for 5 dBZ$_e$":

Not clear what do you mean by 5 dBZe. Does it correspond to the reflectivity that exceeds 5dBZ?

RESPONSE:

Here, we mean the $SD_{diff}$ with an echo intensity of 5 dBZ$_e$.

Thank you very much.

---

## Author Response (AR1)

Dear editor and referees,

We really appreciate the time and effort of you and the reviewers. Your comments definitely help us to improve our paper, and we will make the corrections according to the comments in the revised manuscript.

The following is a point-by-point response to the specific comments from referee 1:

MAJOR COMMENTS
1. TERMNAL VELOCITY AT 94GHz.
The conclusion of the new paper in the abstract lines 21-24 that heavier precipitation occurs in the tropics is to be expected, but it would be useful to know the values of reflectivity and the type of particles that are needed to produce the high terminal velocities above 6 m/s at 94GHz, 7500Hz and consequent folding. In most rainfall Mie scattering of the larger drops at 94GHz leads to terminal velocities much below 6 m/s.

RESPONSE:
In H22, Fig. 9(a) shows a 2D-histogram of $V_{jsim}$ without the random error as a function of the $Z_e$ for the precipitation case. Large fall velocities exceeding 6 m/s are not seen as you pointed out. As shown in Fig. 9(b-d) in H22, considering the random error due to the Doppler broadening, the velocity folding occurs.
We will add the explanation in the revised manuscript.

CHANGE:
We added the following explanation in L230 of the revised manuscript.
This does not mean, however, that the mean Doppler velocity of the precipitation echo exceeds $V_{max}$. In H22, Fig. 9(a) shows a 2D-histogram of $V_{jsim}$ without the random error as a function of the $Z_e$ for the precipitation case. Large fall velocities are not seen due to Mie scattering of the larger drops at 94-GHz. As shown in Fig. 9(b-d) in H22, considering the random error due to the Doppler broadening, the velocity folding occurs.

2. MULTIPLE SCATTERING.
The authors appear to have neglected the effect of multiple scattering which leads to very noisy phase returns due to the differing path lengths of the multiply scattered photons. This effect becomes important for rain rates above 5 mm/hr and will drastically degrade the quality of the Doppler, see Matrosov et al. 2008. https://doi.org/10.1175/2008JTECHA1095.1

RESPONSE:
We thank for your important remarks. Our forward model is based on the single scattering assumption. There are some studies on multiple scattering using Monte Carlo methods (e.g., Matrosov et al. 2008; Battaglia and Tanelli, 2011). Especially the effect of multiple scattering to the Doppler velocity is discussed in Battaglia and Tanelli (2011). In this study, we focus on Doppler errors caused by Doppler broadening and folding, so we do not consider multiple

scattering for simplicity. This issue will be the subject of future research.
We will add this explanation in the revised manuscript.

CHANGE:
We added this explanation in L70 of the revised manuscript and the new references.

Battaglia, A. and Tanelli, S.: DOMUS: DOppler MUltiple-Scattering simulator, IEEE Trans. Geosci. Remote Sens., 49, 442–450, https://doi.org/10.1109/TGRS.2010.2052818, 2011.

Matrosov, S. Y., Battaglia, A., and Rodriguez, P.: Effects of multiple scattering on attenuation-based retrievals of stratiform rainfall from CloudSat, J. Atmos. Ocean. Technol., 25, 2199–2208, https://doi.org/https://doi.org/10.1175/2008JTECHA1095.1, 2008.

3. MODEL RESOLUTION.
The model has a resolution of 3.5km so the size of the features that can be represented is probably greater than 10km. This means that the forward modeled values of reflectivity for each km in the horizontal will not be independent but will be smoothed, and secondly the full range of updrafts and downdrafts will not be resolved. In the current analysis based on the NICAM model it is assumed that any updraft above 3m/s is deemed to be unlikely, but in reality updrafts much higher than this do occur on the km or sub km scale.

RESPONSE:
We note that there may be fast updrafts on the km or sub km scale. However, such events are rare globally and would be negligible in statistics such as latitudinal zonal means. This study focuses on global statistical results and therefore we use the NICAM. When higher horizontal resolution NICAM data becomes available, we would like to study similar evaluation with it. We will add this explanation in the revised manuscript.

CHANGE:
We added this explanation in L79 of the revised manuscript.

1. a) Rather than quoting equation (2) and referring to the HH paper for explanations, the introduction should have a short paragraph explaining that the Doppler is retrieved by estimating the phase change of the returned signal from a target from successive transmitted pulses.

RESPONSE:
We agreed with your suggestion. We will add the following sentence in the revised manuscript. "The EC-CPR measures Doppler velocities using the pulse-pair method. It measures phase shift of echoes from two successive transmitted pulses."

CHANGE:

We added this sentence in L44 of the revised manuscript.

2. b) Include a couple of sentences explaining that as a spacecraft is a moving platform with a finite beam-width the targets have a high Doppler width (4m/s) and so the reshuffling of the targets in the time between two transmitted pulses leads to a rapid lowering of the correlation of the phases and higher Doppler errors.

RESPONSE:
We agreed with your comment. We will add the following sentence in the revised manuscript. "Since the EC-CPR is a finite beamwidth on fast moving spaceborne platform, targets have a broad Doppler width, which causes a worsening of the correlation of the phase. Then, large Doppler errors are introduced."

CHANGE:
We added this sentence in L45 of the revised manuscript.

3. c) The terminology "high mode PRF" (e.g. the caption to figure 3) can be confusing. Does "high" refer to the PRF or the maximum altitude? Better to say high prf /lower maximum altitude.

RESPONSE:
We agreed with your suggestion. It is indeed confusing, so we will change them in the revised manuscript as follows:
"high mode PRF" to "PRF of the high mode (lower PRF)" and
"low mode PRF" to "PRF of the low mode (higher PRF)".

CHANGE:
We change the terminology in L142, L169, L172, L212, L217, L227, L234, L264, L265, L305, L306, L312, L316, L322, L324, L326, L344, and Figure captions in Fig.3-5 of the revised manuscript.

The following is our response to the specific comments from referee 2:

Major comments:
1. The authors assume a constant Doppler spectrum width of 4 m/s. It would be beneficial for readers to explain why this value is a reasonable estimate given the spacecraft speed and the beam-width of the radar. Moreover, it should be also mentioned that the Doppler spectrum width depends on the observed hydrometeors distribution, i.e., for heavy rain the width will be additionally increased.

RESPONSE:
We have omitted this explanation in this paper because we have mentioned it in detail in a previous paper. However, we admit that it would certainly be more helpful to reiterate it as follows:

The width $\sigma_v$ can be considered as a sum of contributions by each. That is,

$$\sigma_v^2 = \sigma_{sm}^2 + \sigma_t^2 + \sigma_{psd}^2,$$

where $\sigma_{sm}$ is the spread due to satellite motion, given by $\sigma_{sm} \sim 0.3 V_{sat} \theta_{3dB}$, $V_{sat}$ is the satellite velocity, and $\theta_{3dB}$ is the beam width (Sloss and Atlas 1968). When $V_{sat}$ is 7738 m/s and $\theta_{3dB}$ is 0.00166 rad (0.095°), $\sigma_{sm}$ becomes 3.85 m/s. The spread $\sigma_t$ is due to turbulence and $\sigma_{psd}$ to the distributions of hydrometeor falling velocities, respectively, which are assumed to be $\sigma_t = 1.0$ m/s (Amayenc et al., 1993), and $\sigma_{psd} = 0.5$ m/s (Gossard et al., 1997). As for the latter term, it is reported to spread to 1.0 m/s for rain (Lhermitte 1963). In this study, we assumed the $\sigma_{psd,} = 0.5$ m/s so that $\sigma_v$ becomes 4.01 m/s.

We will add our explanation in the revised manuscript.

CHANGE:
We added our explanation in L125 and the new references.

Sloss, P. W. and Atlas, D.: Wind shear and reflectivity gradient effects on Doppler radar spectra, J. Atmos. Sci., 25, 1080–1089, 1968.

Amayenc, P., Testud, J., and Marzoug, M.: Proposal for a Spaceborne Dual-Beam Rain Radar with Doppler Capability, J. Atmos. Ocean. Technol., 10, 262–276, https://doi.org/10.1175/1520-0426(1993)010<0262:PFASDB>2.0.CO;2, 2002.

Gossard, E. E., Snider, J. B., Clothiaux, E. E., Martner, B., Gibson, J. S., Kropfli, R. A., and Frisch, A. S.: The potential of 8-mm radars for remotely sensing cloud drop size distributions, J. Atmos. Ocean. Technol., 14, 76–87, https://doi.org/10.1175/1520-0426(1997)014<0076:TPOMRF>2.0.CO;2, 1997.

Lhermitte, R. M.: Motions of scatterers and the variance of the mean intensity of weather radar signals, Rep. SRRC-RR-63-57, Sperry Rand Research Center, 43 pp., 1963.

2. Although, a long integration path (10 km) seems to be a tempting approach to reduce uncertainty in the Doppler measurements, the authors do not assess the effect of a long scale signal decorrelation and non-uniform beam filling effects in such a large sampling volume. Moreover, the horizontal resolution of 10 km prevents studies on the small scale features like localized convection, the characterization of which is one of the objectives of space-borne Doppler radar missions.

RESPONSE:
We have already evaluated the change in Doppler error for 500 m, 1 km, and 10 km horizontal

integrations in H22. It has been demonstrated that the Doppler error was significantly reduced by the 10-km integration. It indicated that the reduction of random errors by the integration had a larger contribution compared to the error by small-scale Doppler changes. Therefore, this paper mainly investigates the results of the 10-km integration.

3. Does the radar simulator accounts for multiple scattering? This issue has been demonstrated to have a destructive effect on the quality of the Doppler measurements.

RESPONSE:
We thank for your important remarks. Our forward model is based on the single scattering assumption. There are some studies on multiple scattering using Monte Carlo methods (e.g., Matrosov et al. 2008, Battaglia and Tanelli, 2011). Especially the effect of multiple scattering to the Doppler velocity is discussed in Battaglia and Tanelli (2011). In this study, we focus on Doppler errors caused by Doppler broadening and folding, so we do not consider multiple scattering for simplicity. This issue will be the subject of future research.
We will add this explanation in the revised manuscript.

CHANGE:
We added this explanation in L70 of the revised manuscript and the new references.

Battaglia, A. and Tanelli, S.: DOMUS: DOppler MUltiple-Scattering simulator, IEEE Trans. Geosci. Remote Sens., 49, 442–450, https://doi.org/10.1109/TGRS.2010.2052818, 2011.

Matrosov, S. Y., Battaglia, A., and Rodriguez, P.: Effects of multiple scattering on attenuation-based retrievals of stratiform rainfall from CloudSat, J. Atmos. Ocean. Technol., 25, 2199–2208, https://doi.org/https://doi.org/10.1175/2008JTECHA1095.1, 2008.

Minor comments:
1. The terminology "high-mode PRF" can be misleading. It would be better to use "low PRF mode" or "high tropopause mode".

RESPONSE:
We agree with your suggestion. It is indeed confusing, so we will change them in the revised manuscript as follows:
"high mode PRF" to "PRF of the high mode (lower PRF)" and
"low mode PRF" to "PRF of the low mode (higher PRF)".

CHANGE:
We change the terminology in L142, L169, L172, L212, L217, L227, L234, L264, L265, L305, L306, L312, L316, L322, L324, L326, L344, and Figure captions in Fig.3-5 of the revised manuscript.

2. It feels like some of the discussions can be reduced in length, e.g., when the high PRF mode is compared with the low PRF mode.

RESPONSE:
We thank you for your remarks. The main progress of this paper is that we extended the analysis to 16 orbits and also performed low mode PRF analysis and compared the results with those of high mode PRF. For the latter, we first discuss the high mode PRF results, and in the sections where we discuss the comparison with low mode PRF, we have tried to avoid redundancy and only mention what is necessary. If you feel that the discussion is redundant, it may be due to the difficulty of reading owing to the terminology problem mentioned above. We will change them in the revised manuscript in accordance with the points mentioned above, and we believe that the difficulty of reading has been improved.

Some minor comments are also included in the attached file.

Page 1,
Line 16, "mean Doppler errors for 5 dBZe":
Not clear what does it mean

RESPONSE:
Here, we mean the average Doppler errors with an echo intensity of 5dBZe.

Page 3,
Line 74 "threshold $\geq$20":
Is this a threshold on the reflectivity or on the cloud mask product? in the latter case, what does this number correspond to?

RESPONSE:
This is the threshold for the cloud mask product and corresponds to the "weak echo" in this product (Marchand et al., 2008). This threshold is used in many other CloudSat-based hydrometeor studies (e.g., Sassen & Wang 2008).
We will add our explanation in the revised manuscript.

CHANGE:
We changed our description for clarity in L86 of the revised manuscript as follows as well as added the new references:
(before)
For the observed data, we used the CPR Level 2B-GEOPROF cloud mask product to extract bins with threshold $\geq$20 that are less affected by surface clutter and other factors. These are estimated to have a false-positive probability of 5 % (Marchand et al., 2008).

(after)

For the observed data, we defined the hydrometeor bin as where the cloud mask value is greater or equal than 20 from the CPR Level 2B-GEOPROF product, which means a weak, good, or strong echo detection (Marchand et al., 2008). These are estimated to gives an estimated false detection rate smaller than 5 %. This value is adopted in many other CloudSat-based hydrometeor studies (e.g., Sassen & Wang 2008).

Sassen, K. and Wang, Z.: Classifying clouds around the globe with the CloudSat radar: 1-year of results, Geophys. Res. Lett., 35, https://doi.org/10.1029/2007GL032591, 2008.

Line 75: insert "of cloud occurrence"

RESPONSE:
We will change our statement in the revised manuscript accordingly.

CHANGE:
We changed our statement in L89 of the revised manuscript accordingly.

Page 7,
Line 145: "Incidentally, we added black dashed and solid lines to Fig. 3 to show the results":
Not clear. These figures have 4 lines that were described before. Please, rephrase this sentence.

RESPONSE:
We will correct in the revised manuscript as follows:
(before)
Incidentally, we added black dashed and solid lines to Fig. 3 to show the results.

(after)
Note the black dashed and solid lines in Fig. 3.

CHANGE:
We corrected our statement in L204 of the revised manuscript.

Line 146: insert "in H22"?

RESPONSE:
We will also change our statement in the revised manuscript as follows:
(before)
In both Figs. 3a and 3b, the results are in good agreement with those of this study.
(after)
In both Figs. 3a and 3b, the results in H22 are in good agreement with those of this study.

CHANGE:
We changed our statement in L204 of the revised manuscript.

Line 148: "The PRF varies from 7156 to 7500 Hz, with a corresponding SDrandom of 1.5 to 3.4 for 0 to –19 dBZe (see Fig. 2 in H22). On the other hand, in the high mode, the PRF varies from 6106 to 6464 Hz, with a corresponding SDrandom of 0.8 to 1.5 for 0 to –19 dBZe":
SD_random is lower for higher PRF.

RESPONSE:
We will correct in the revised manuscript as follows:
(before)
The PRF varies from 7156 to 7500 Hz, with a corresponding $SD_{random}$ of 1.5 to 3.4 for 0 to –19 dBZe (see Fig. 2 in H22). On the other hand, in the high mode, the PRF varies from 6106 to 6464 Hz, with a corresponding $SD_{random}$ of 0.8 to 1.5 for 0 to –19 dBZe.

(after)
The PRF varies from 7156 to 7500 Hz, with a corresponding $SD_{random}$ of 0.8 to 1.5 for 0 to –19 dBZe (see Fig. 2 in H22). On the other hand, in the high mode, the PRF varies from 6106 to 6464 Hz, with a corresponding $SD_{random}$ of 1.5 to 3.4 for 0 to –19 dBZe.

CHANGE:
We corrected our statement in L208-209 of the revised manuscript.

Page 8,
Line 167: "the Doppler accuracy should be higher in the tropics and lower toward the poles.":
In the tropics, high mode will be used (according to your description in line 89) that is characterized by lower PRF thus higher SD_random which contradicts what is written here.

RESONSE:
The statement in line 89 merely states what mode is used in nominal operation. However, for clarity, we will change our statement in the revised manuscript as follows:
(before)
From the PRF variation shown in Fig. 2, the Doppler accuracy should be higher in the tropics and lower toward the poles.

(after)
From the PRF variation shown in Fig. 2, in the PRF of the high mode (lower PRF), the Doppler accuracy should be higher in the tropics and lower toward the poles.

CHANGE:
We changed our statement in L227 of the revised manuscript.

Line 169: "the frequency of precipitation echoes is considered to be the highest in the tropics, and the resulting folding Doppler error may have resulted in the largest SDdiff being in the tropics.":
please rephrase.

RESPONSE:
We will correct our statement in the revised manuscript as follows:
(before)
the frequency of precipitation echoes is considered to be the highest in the tropics, and the resulting folding Doppler error may have resulted in the largest $SD_{\text{diff}}$ being in the tropics.

(after)
the frequency of precipitation echoes is considered to be the highest in the tropics, and the folding Doppler error may have resulted in the largest $SD_{\text{diff}}$ in the tropics.

CHANGE:
We corrected our statement in L229-230 of the revised manuscript.

Line 174: "This may be related to the frequency of precipitation echoes":
Rather than the frequency, it is the intensity of the precipitation that matters. The strongest radar signal is expected to be observed for the most intense rain events that are characterized by the largest raindrops thus the largest terminal and Doppler velocities. Of course, the attenuation due to rain will result in a steep reflectivity decrease toward the ground but this does not change the fact that velocity folding occurs for the most intense rain.

RESPONSE:
In H22, Fig. 9(d) shows a 2D-histogram of the Doppler velocities after 10 km integration as a function of the Ze. The velocity folding occurs even for 0 dBZe, indicating that the heavy rain with large Ze does not make a significant contribution. Therefore, we consider the frequency of precipitation to be more important statistically.

Line 177 & 179: "for 5 dBZe":
Not clear what do you mean by 5 dBZe. Does it correspond to the reflectivity that exceeds 5dBZ?

RESPONSE:
Here, we mean the SDdiff with an echo intensity of 5 dBZe.

Page 9,

According to your description, high-PRF mode will be used for latitudes beyond 60 degrees and low-PRF mode for altitudes below 60 deg. Therefore, half of the presented panels are not necessary. Please consider removing them or make a note in the figure caption which mode is expected to be used in which zone.

RESPONSE:
As already mentioned, the description in line 89 only states what mode is used in nominal operation.

Page 10, "for 5 dBZe":
Figure 6, caption & Line 196: Not clear what do you mean by 5 dBZe. Does it correspond to the reflectivity that exceeds 5dBZ?

RESPONSE:
Here, we mean the values with an echo intensity of 5 dBZe.

Line 201: "due to the frequency of precipitation echoes":
As previously, I don't think it is the frequency of occurrence that matters but the rainfall intensity.

RESPONSE:
As stated in the reply to Line 174, we consider the frequency of precipitation to be more important statistically.

Line 201: "there would be no relationship between the latitudinal variation of SDdiff with unfolding correction and the frequency of precipitation echoes":
That is true only if only high SNR data are considered. The SD_diff depends on the SNR, thus the regions where weak radar echoes are observed often are characterized by higher SD_diff.

RESPONSE:
The latitudinal variation of SDdiff with unfolding correction for -19dBZe with relatively low SNR is shown below. As in fig. 6, there is no significant variation in latitude.

[Figure]

Page 12,
Line 242: "for 5 dBZe":
Not clear what do you mean by 5 dBZe. Does it correspond to the reflectivity that exceeds 5dBZ?

RESPONSE:
Here, we mean the SDdiff with an echo intensity of 5 dBZe.

Thank you very much.

---

## Author Response (AR2)

Dear editor,

We really appreciate the time and effort of you. Your comments definitely help us to improve our paper, and we will make the corrections according to the comments in the revised manuscript.

The following is a point-by-point response to the specific comments:

At four locations in the paper Reviewer #2 requests an explanation for "mean Doppler errors for 5 dBZe" and other similar text. Please edit the text at these locations to provide clarification, not simply to say that you mean that you select only situations where the echo intensity is 5 dBZe, but to explain whether you are referring to the mean error (i.e. the bias) in Doppler velocity, or the root-mean-squared error in Doppler velocity. Similarly for SDdiff.

RESPONSE:
We admit that our statement was unclear. We changed our statements in L16, L24, L53, L59, L152, L154, L158, L159, L181, L186, L192, L212, L265, L268, L275, L277, L300, L301, and L303 of the revised manuscript.

At one point you added the line "Note the black dashed and solid lines in Fig. 3." which is meaningless: what is the reader supposed to note about these lines? Please state what the black lines show.

RESPONSE:
We changed our statement as below in L179 of the revised manuscript.

"The black lines in Fig. 3 are the result for H22, the dashed lines denote the $SD_{\text{diff}}$, and the lines indicate the $SD_{\text{diff}}$ with unfolding correction (using the same method as in Eq. (7)). "

The data availability statement says the data are available by email. Please place the data in a public repository and refer to that instead. Perhaps the data are the same as https://doi.org/10.5281/zenodo.7835229, in which case please cite that (Roh et al. 2023).

RESPONSE:
Accordingly, we changed our statement and added the following information in the "Data availability" section.

"The NICAM/J-Sim data with two orbits is available in the repository (https://doi.org/10.5281/zenodo.7835229). We also will give a full data set if you request."

Thank you very much.